# An Overview of the Non-Energetic Valorization Possibilities of Plastic Waste via Thermochemical Processes

**DOI:** 10.3390/ma17071460

**Published:** 2024-03-22

**Authors:** Kazem Moussa, Sary Awad, Patricia Krawczak, Ahmad Al Takash, Jalal Faraj, Mahmoud Khaled

**Affiliations:** 1Energy and Thermo-Fluid Group, Lebanese International University, LIU, Bekaa P.O. Box 146404, Lebanon; kazem.h.moussa@gmail.com (K.M.); ahmad.takash@liu.edu.lb (A.A.T.); jalal.faraj@liu.edu.lb (J.F.); mahmoud.khaled@liu.edu.lb (M.K.); 2IMT Atlantique, GEPEA, UMR CNRS 6144, 4 Rue Alfred Kastler, F-44000 Nantes, France; 3IMT Nord Europe, Institut Mines-Télécom, University of Lille, Centre for Materials and Processes, 941 rue Charles Bourseul, CS 10838, F-59508 Douai, France; patricia.krawczak@imt-nord-europe.fr; 4Energy and Thermo-Fluid Group, The International University of Beirut BIU, Beirut P.O. Box 146404, Lebanon; 5Center for Sustainable Energy & Economic Development (SEED), Gulf University for Science & Technology, Hawally P.O. Box 7207, Kuwait

**Keywords:** plastic waste, chemical recycling, upcycling, pyrolysis, chemical feedstock

## Abstract

The recovery and recycling/upcycling of plastics and polymer-based materials is needed in order to reduce plastic waste accumulated over decades. Mechanical recycling processes have made a great contribution to the circularity of plastic materials, contributing to 99% of recycled thermoplastics. Challenges facing this family of processes limit its outreach to 30% of plastic waste. Complementary pathways are needed to increase recycling rates. Chemical processes have the advantage of decomposing plastics into a variety of hydrocarbons that can cover a wide range of applications, such as monomers, lubricants, phase change materials, solvents, BTX (benzene, toluene, xylene), etc. The aim of the present work is to shed light on different chemical recycling pathways, with a special focus on thermochemicals. The study will cover the effects of feedstock, operating conditions, and processes used on the final products. Then, it will attempt to correlate these final products to some petrochemical feedstock being used today on a large scale.

## 1. Introduction

The world is being affected by the creation of new technologies and an increase in production levels in various industrial fields, thus raising questions about environmental and ethical issues relating to sustainability. Studies have also shown that a corporation’s social practices (including sustainable practices) affect the customers’ commitment, so it is not only an environmental issue but also an economic one [1].

Regarding the plastics industry, it is now undoubtedly clear that the rise of polymeric material production and use is exponential, making the implementation of reduction, reuse, repurpose, and recycling strategies crucial. Figure 1 shows the drastic yearly jumps in plastic production since the 1950s to the year 2021. In the year 1950, the world produced 1.5 million metric tons of plastic, compared to 390.7 million metric tons produced in the year 2021 [2]. Because there are currently no other alternatives to plastic in a number of applications, these numbers are expected to increase at higher rates in the years to come. In 2021, China produced around 32% of the world’s plastics, while North America produced 18% and Europe produced only 15% in the same time period [2].

The various types of plastics that are currently being used in the manufacturing industry are categorized into seven main families: polypropylene (PP), high-density polyethylene (HDPE), low-density polyethylene (LDPE), polyvinyl chloride (PVC), polyethylene terephthalate (PET), polystyrene (PS), and “others” [3]. It is worth keeping in mind that the “others” category includes a wide range of very different engineering and specialty polymers. As seen in the graph in Figure 2, the overall global production of plastic is increasing, but the rate also depends on the type of polymer. For example, over the period 2020–2050, polystyrene, polyethylene terephthalate, and polyvinyl chloride values are not expected to show excessive jumps. On the other hand, it can be observed that the other categories in the list are going to cause more environmental issues, especially the “others” group, where production is expected to reach 171 million metric tons by 2050, while polypropylene production will jump to 107 million metric tons [4].

Due to the high rate of increase in the production of plastics in various fields, such as packaging, automotive, construction, and household/consumer-related products, it is predicted that large quantities of plastic waste will be produced. Figure 3 illustrates these wastes and categorizes them based on their field of industry and application over the years from the 1980s to 2019. As can be seen, the packaging industry has a much higher increase rate than the other applications, with up to 142 million metric tons of waste released in 2019 (~40%), while the vehicle industry released 35 million metric tons in the same year.

This general trend in plastic waste production raises important questions regarding its sustainability and effects on the environment and human health. Thus, the key players in the plastics value chain have become aware, at least in Europe, that it is urgent to evolve towards a circular, low-carbon footprint plastics industry [5].

**Figure 3 materials-17-01460-f003:**
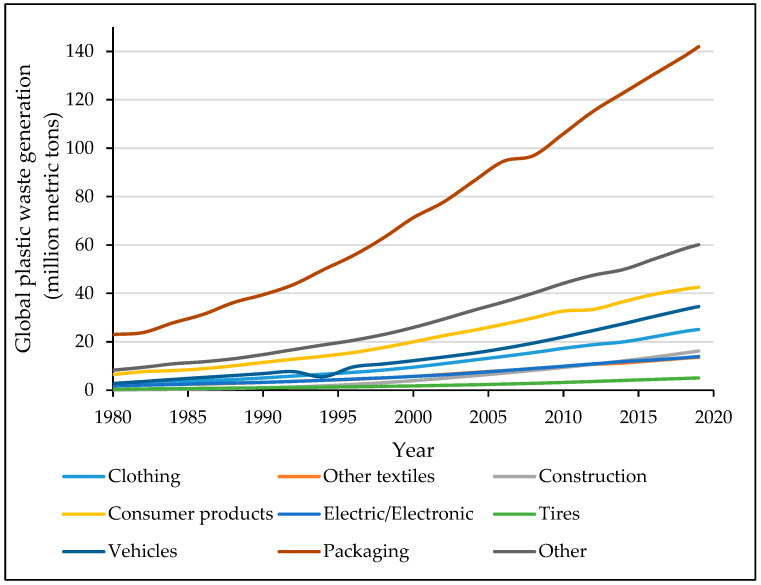
Global plastic waste generation per application from 1980 to 2019 [6].

PlasticsEurope, the leading European trade association, has commissioned the report ‘’ReShaping Plastics: Pathways to a Circular, Climate-Neutral Plastics System in Europe’’ [7], which includes recommendations for transitioning to the EU’s net-zero carbon emissions and circularity goals by 2050. Subsequently, their ‘’Plastics Transition’’ roadmap [8] was issued, providing a framework, milestones, and indicators to monitor progress, identify bottlenecks, and find solutions for making plastics circular, thus driving the plastics life cycle to net-zero emissions and fostering sustainable use of plastics. Plastic waste recycling and valorization will contribute to achieving these goals.

Traditional recycling methods (mechanical recycling) of plastic waste have limitations due to the fact that they cannot break down polymer chemical bonds [9]. Their purpose is to melt and reshape the polymer; thus, they cannot correct the aging of the plastic and its structural damages, which leads to a downcycling system. The maximum achieved recycling rate can reach 31%, and in this condition, the carbon footprint share of plastics can decrease by one quarter to 3.4% [10].

On the other hand, chemical processing can be a complementary solution that can upcycle plastic waste by breaking down complex polymers and mixtures of plastic waste into simpler elements, such as hydrocarbons, hydrogen, syngas, and/or carbonaceous materials. These simple structure elements have essential roles in production processes and industrial applications and will be analyzed in this paper. Each process has its own operating conditions and respective products depending on the feedstock.

The present work will highlight the global issues of plastic waste, including its production levels and the seriousness of its effects. Then, the main issues hindering the mechanical recycling of these materials and complementary solutions will be discussed, with a focus on the pyrolysis process, as it shows a good compromise between feedstock selectivity, process complexity, and product variety. On the other hand, the product families that can be obtained by these processes and their possible uses as chemical feedstock will also be discussed. Finally, the challenges facing chemical recycling (with a focus on the pyrolysis process) will be presented.

## 2. Challenges Facing Plastic Recycling

Of the cumulated plastic waste produced since 1950, only 9% has been recycled and 22% has been mismanaged [11]. These values call for immediate actions regarding the rise of plastic waste to find recycling processes and valorization possibilities for them.

Several issues are leading to the low rate of recycling and will be discussed in this section. Actually, 99% of recycled thermoplastics (representing 80% of total plastics) are obtained through mechanical recycling. This mode of recycling is very demanding in terms of the product’s purity and capital and operating expenditures, and it is limited due to the downcycling effect related to the degradation of recycled plastic quality along cycles. On the other hand, composite plastics and thermoset materials cannot be molten down and reshaped; thus, they should undergo other forms of valorization [12].

### 2.1. Quality of Plastic Waste

Plastic waste can be classified into two categories in terms of origin: industrial waste plastics and post-consumer waste plastics. The former is well sorted and clean and requires minimum pretreatment before recycling. The latter is more challenging because it is a mixture of plastics with different qualities, meaning it is very heterogeneous in terms of resins and contains high amounts of contaminants. Furthermore, within the post-consumer category, wastes of the same polymer can have different qualities related to their uses and lifecycle (degradation because of UV, oxidation, fatigue, wear, etc.). At the same time, these plastics could contain different additives that match their end-use purposes. The diversity and complexity of these additives are illustrated in Table 1, where it can be seen that the presence of metals and chemicals can change the final aspect of the mixture and its compatibility with different end-uses of the resulting recycled plastic.

As the quality of the final recycled plastic is a result of the mixture quality and additive content, its use can be restricted to domains related to highly resistant plastics and food, medical, and pharmaceutical industry applications.

### 2.2. Purity

Mechanical recycling is not intended to change the chemical structure of plastics; thus, it is dependent on the feedstock quality. Differences in the degradation temperature and the low miscibility of molten plastics can lead to problems in the recycling process and the quality of the final product. Thus, this technique requires feedstock purity to be higher than 97% [18], which is ensured by a series of sorting processes. These processes include a combination of manual and automated processes [19] that are costly in terms of labor time (manual sorting and flotation) and capital investment (spectroscopic and laser-based separation) and can induce water treatment and energy-intensive issues (floatation separation). The common efficiency of separation can barely exceed 50%, which means that around half of the plastic stream entering the process exits it in the form of mixed non-sortable plastics. In these mixtures of non-sortable plastics, we can find some colored plastics [20], some others where additives interfere with spectroscopic identification techniques [21], and multilayered and blended plastics [22]. The latter are blended in order to reach specific thermal and mechanical properties that cannot be obtained with single resins. They can be either miscible or immiscible and can cover a wide range of combinations, as illustrated in Table 2.

## 3. Alternative Plastic Waste Treatment Processes

Chemical recycling can be a good solution to increase the recycling rate of waste that cannot undergo mechanical recycling. In general (except for extraction and dissolution processes), chemical recycling acts on the chemical bonds inside the feedstock in order to depolymerize it. The depolymerization can be total, meaning monomers can be recovered to serve as feedstock for a new polymerization cycle, but it can also be partial, where general feedstock for the chemical industry or alternative fuels can be obtained. Furthermore, in the case of gasification and steam reforming, the feedstock can be transformed into a gas with a high hydrogen content. Finally, we can cite the processes intended to produce hydrogen and carbon materials from plastics, as in the case of microwave-assisted catalytic pyrolysis [29] or flash Joule heating [30]. In this section, the main chemical treatment processes will be presented, with a focus on pyrolysis.

### 3.1. Solvent-Based Processes

Depending on their nature and additive content, polymers can undergo different chemical processes. Extraction, for example, is used when additives have to be removed from plastic waste. In this case, a solvent has a good affinity for the targeted additive, a low affinity for the polymer itself, and a good ability to swell the fibers so the additive molecules can be removed. On the contrary, dissolution aims to use a solvent with a high affinity for the polymer matrix itself in order to dissolve it (without additives). Then, pure polymers can be recovered using an antisolvent that leads to their precipitation. The solvent and the antisolvent are recycled at the end of the reaction to be used on other cycles. These physical techniques use large amounts of chemicals and high energy in order to evaporate and recover solvents. Furthermore, the toxicity of solvents and the complexity of additives that can be found in mixed plastic matrices limit the uses of these techniques to specific cases.

The use of solvents and additives in the making of plastic mixtures increases the risk of chemical toxicity and poses some dangers in their thermochemical recycling processes. These components can affect human life as well as the environment [31]. Phthalates are used as plasticizers and can cause intoxication upon exposure through injection and inhalation [32]. Bisphenol A (BPA) and its substitutes, such as bisphenol S (BPS) and diphenyl sulfone, are also toxic additives added to many plastic mixtures [33]. They can impact reproductive system development and function in male rodents [34].

### 3.2. Depolymerization Processes

Before choosing the depolymerization technique, polymerization should be explained. There are two main general methods for producing polymers: addition polymerization and condensation polymerization.

The synthesis of polymers using addition polymerization can be conducted using four techniques: radical, cationic, anionic, and coordination catalytic polymerization. In radical polymerization, a radical initiator is broken down thermally, and then a radical attacks the π-bond in the alkene monomer, creating a covalent bond with one of the carbon atoms and turning the other atom into a reactive radical. This step repeats and propagates until the process terminates when the radical chains combine or are neutralized. A visual representation can be seen in Figure 4 [35]. Cationic and anionic polymerization go through the same steps; however, the initiator will have to be strong acids or bases to convert the alkene monomer into an anion or a cation. The fourth method is catalytic polymerization, which involves the Ziegler–Natta catalyst, which gives more control over the structure of the produced polymer and makes it more feasible to reach higher molecular weights at lower pressure and temperature levels.

On the other hand, condensation polymerization consists of connecting two bi-functional monomers so that each one of them can connect with two other monomers [36].

Many plastic materials are produced through this technique, like polyethylene terephthalate (PET), polycarbonate, and nylon. Figure 5 shows how di-carboxylic acid and 1,6-hexamethylenediamine, each containing two functional groups (-OH and -NH_2_), are brought to high temperatures and react together to form a chain of polymers through a slow reaction [37].

In the addition polymerization family, we can find polyolefins, polystyrene, and PVC. They usually have high bond energy, which requires high amounts of energy in order to depolymerize them. PMMA and PS are an exception to this rule as they have side groups that fragilize the C-C bond in the beta position of the branching. On the other hand, condensation polymers have lower bond energies and can contain polar functions that allow attack with specific solvents (solvolysis) or chemicals (transesterification and transamidation). In general, the lower the carbon bond energy, the lower the depolymerization energy, and thus the production of monomers is recommended. On the other hand, general feedstock production is recommended for feedstock with higher depolymerization energy.

#### 3.2.1. Solvolysis

Solvolysis is a process where the organic matter is degraded (depolymerized in the case of plastics) under the effect of a hot pressurized solvent (maintained in a liquid state). The effect of the solvent lowers the temperature at which depolymerization occurs, either by acting as a solvent or as a reagent (e.g., supercritical alcohol transesterification). These reactions can be accelerated using acid or base catalysts. The operating conditions reported in some research works are listed in Table 3. This process can also be used in the recycling of composite materials by selectively decomposing the matrix or the fiber component. The solvolysis still requires a minimum quality of feedstock and faces challenges regarding the extraction of monomers from the reactive medium (solvents, catalysts, oligomers, and contaminants coming from the feedstock) [39,40].

#### 3.2.2. Pyrolysis and Gasification

##### Liquid-Oriented Processes

Pyrolysis is a thermochemical process that degrades polymers into separate solid, liquid, and gas materials under the effect of high-temperature (above decaying temperature) heating in the absence of oxygen. This process decays several bonds in the polymers and their additives. Lots of studies have dealt with the pyrolysis of polyolefins to produce a large range of products. It can be performed at a wide range of temperatures (300–900 °C) [46] in the presence or absence of catalysts. The first is called thermal pyrolysis. It is mainly based on random scission cracking and produces linear hydrocarbons. The products can range from heavy waxes at the lowest temperatures to olefin-rich gases at the highest temperatures. Usually, polyolefin cracking produces very low amounts of char. On the other hand, catalytic cracking can produce a large range of hydrocarbons depending on the type and selectivity of the catalyst. Acid catalysts, mainly zeolites, are the most commonly used catalysts. Pyrolysis using acid catalysts follows a carbenium-ion mechanism, where branched and aromatic hydrocarbons are abundantly produced. The range of carbon chain lengths and the amounts of aromatics are mainly dependent on the type (Lewis or Bronsted) of acid sites, their strength and density, and the pore size of catalysts. For example, the ZSM-5 zeolite is known for its production of high amounts of aromatics and light hydrocarbon gases due to its microporous nature and the high density of its acid sites [47]. Y-type catalysts (USY) are known for their low aromatic production and high-branched product shares [48,49] due to their mesoporosity and moderate acidity. Besides synthetic zeolites, natural zeolites and clay-based catalysts are gaining more attention due to their lower costs. Tang et al. [50] investigated the effects of metakaolin-based geopolymers on the cracking of low-density polyethylene (LDPE). They obtained a liquid yield of 88%, with the majority in the diesel (81%) and gasoline (10%) ranges. Miandad et al. [51] used natural zeolites in the pyrolysis of four types of polymers (PP, PE, PS, and PET) at 550 °C. Polyolefin cracking on acid-activated natural zeolites led to liquid yields of 42 and 54%, mainly in the diesel fuel range.

Besides polyolefines, pyrolysis has proven to be very efficient in the depolymerization of PMMA, PS, and nylon-6 to produce their respective monomers. In the case of PS, the stereo-hindrance effect of benzene makes the monomer bonds strong enough to create a preferential cracking pathway that occurs on the bonds between monomers [52]. In a similar way, the tertiary carbon of PMMA creates a strong bond between it and its β-carbon, which leads to a preferential scheme at the junction between two monomers. Finally, for nylon-6, the weak bond energy on the polar edges of the monomer also creates a preferential mechanism of thermal cracking. The depolymerization efficiencies of these polymers can exceed 75% and reach up to 98% under pyrolysis conditions at a lab scale [53]. At an industrial level, Agilyx in the US has succeeded in achieving a styrene recovery rate of 70% from post-consumer polystyrene in their plant, which has a capacity of 10 tons per day [54]. As with solvolysis, the main challenges facing the depolymerization of these feedstocks via pyrolysis are the separation of monomers from the reaction products and the impurities contained in the feedstock.

On the other hand, PVC and PET are feedstocks that are less compatible with pyrolysis. PVC has the same structure as polyethylene, except that one atom of hydrogen in each monomer is substituted by one atom of chlorine. The dehydrochlorination starts at the early stages of the reaction (220–350 °C) and produces HCl, which is very corrosive for equipment and can contaminate the remaining pyrolysis products. Some solutions have been suggested to deal with this feedstock, such as adding calcium carbonate to the feedstock so it can neutralize the hydrochloric acid or separating the reaction into two steps, with the first one dedicated to the dehydrochlorination and the second one to cracking. In this case, HCl can be stripped and recovered as a byproduct of the process, as in the case of the Klean Industries process developed in Japan, where the patented technology allows the pyrolysis of plastic feedstock containing up to 20% of PVC and reduces chlorine content in the pyrolysis oil below 100 ppm. All other solutions on the market cannot tolerate more than 1% of PVC in the feedstock. When it comes to PET pyrolysis, its main volatile products are waxy and corrosive and mainly composed of benzoic acid, terephthalic acid, and vinyl compounds [55]. These compounds are corrosive for equipment, can stick and clog the piping of systems, and increase the acidity of pyrolysis oils. Furthermore, high char yields are expected from PET pyrolysis due to the competition between aromatic condensation and depolymerization (scission). Thus, the presence of PET in mixed plastics should be avoided.

Table 4 presents examples of the pyrolysis of some plastic waste, along with their operating conditions and their respective products. Each product can be utilized as chemical feedstock, as explained in the next sections.

In recent years, the conversion of polyolefins and plastic mixtures was mainly focused on the production of liquid fuels [49], while the gaseous fraction was meant to feed the system with heat [22] with some focus on the utilization of char byproducts. Recently, there has been more work focusing on chemical feedstock and closed-loop recycling of plastics [63].

In order to obtain targeted products, two possible ways can be observed. The first one consists of using selective catalysts with tuned operating conditions, while the second one consists of refining the pyrolytic oil obtained using different techniques such as fractionated distillation, hydrotreating, or steam cracking. Actually, the most applied technique in industrial plastic pyrolysis plants is distillation, followed by some chemical or physical treatments (decoloring, deodorizing, extraction, etc.).

The first option is more applicable to a single plastic or a family of plastics, while the second is more dedicated to plastic mixtures.

On an industrial scale, most of the applications use thermal pyrolysis followed by a refining scheme. Klean Industries, for example [64], produces pyrolytic liquids that, after refining, can be split into three fractions: a light one that serves as feedstock for recycling, an intermediate one that serves as diesel-like fuel, and a heavy one that is used on site to feed the system with electricity. Plastic Energy, on the other hand, has implemented its technology in Spain, where it is able to treat 30 tons per day of mixed domestic plastic waste and transform the resulting pyrolysis oil into naphta and diesel fuel [65]. One can also cite Alterra Energy in the US, which transforms 60 tons per day of non-hazardous mixed plastic waste into four fractions of what it calls ecofuels [66].

On the other hand, few companies are working on recycling, such as Klean Industries, where a small fraction of its pyrolysis oil is dedicated to producing new plastic feedstock (although the majority of its production is used as fuel). Recycling technologies transform mixed plastic waste into a low-sulfur pyrolytic oil called Plaxx, which is refined in order to produce virgin plastics [67]. Finally, we can cite the recent collaboration between Plastic Energy and ExxonMobil to build an advanced recycling plant in France that is able to convert 25,000 tons per year of waste plastics into virgin-quality polymers after upgrading the pyrolytic oils produced with their technology.

Table 5 illustrates some industrial plants, the technologies used, and the final target products.

##### Carbonaceous Materials and Gaseous Product-Oriented Processes

Gasification

Gasification consists of the transformation of organic matter into a synthesis gas under high temperatures (700–1500 °C) [72] in the presence of a gasifying agent that could be air, oxygen, steam, or a mixture of them. When air and oxygen are used, partial oxidation occurs, leading to an auto-thermal operation of the gasifier. In the case of steam gasification, the endothermal reaction produces higher amounts of hydrogen at the expense of higher tar yields and energy consumption [73]. Plastic waste is commonly gasified alone or (more commonly) in mixtures with other solid wastes and/or biomass. The produced syngas can be used as an energy carrier that can be used to produce heat, in CHP applications [74], or in fuel cells [75]. However, with higher-quality syngas (ultralow tar content and a high H_2_/CO ratio), the production of light hydrocarbons that could serve as chemical or new plastic feedstock will be possible using the Fischer–Tropsch process [72]. Besides Fischer–Tropsch applications, gasification can also be used to produce H_2_ that should be separated from the syngas [76]. The presence of plastics in the feed increases the H_2_/CO ratio, which is more favorable for advanced uses, but on the other hand, it increases the tar content. The tar issue is more prominent when PET and PS are present in the feed [72,73].

The gasification process itself has reached industrial maturity; however, tar and gas separation is still an important issue to be solved before suggesting viable chemical and material recovery applications. In fact, most targeted applications using gasification target the energy and fuel sectors. Advanced applications using Fischer–Tropsh require high-quality syngas and should target larger-scale applications, which can limit material-oriented applications.

Combined Carbonaceous Materials and Hydrogen-Production-Oriented Processes

Solid pyrolysis products can be byproducts of a process that is intended to produce liquid and gaseous hydrocarbons, but they can also be the target product of the process. When the solid phase is the target, a wide range of carbonaceous materials can be produced, such as graphene, carbon dots, carbon nanotubes (CNTs), activated carbon, or other porous carbon-based materials [77], using different types of processes.

These materials can be used in a wide spectrum of applications, such as battery components (anodes, cathodes, or separators) [78,79,80], supercapacitors [81], adsorbents for different applications (water treatment, gas separation, and air pollutants removal) [82], [83], catalysts [84], photocatalysis [85], and composite materials [86].

These carbonaceous materials can be produced via several techniques, among which we can cite slow carbonization, catalytic carbonization, flash Joule heating (FJH) [86], microwave-assisted carbonization [29], and pyrolysis followed by chemical vapor deposition [87].

Among these techniques, FJH and microwave-assisted catalytic carbonization have high potential due to the possibility of simultaneous production of hydrogen and carbonaceous materials. Recently, Jie et al. [29] applied microwave-assisted cracking to different plastic feedstocks using iron-based catalysts in order to produce multiwalled nanotubes and hydrogen. With this technique, they succeeded in extracting 97% of the hydrogen content of plastics. In another study, Wyss et al. [30] applied an FJH process to produce hydrogen and graphene from different types of plastics, with hydrogen extraction efficiencies exceeding 80%.

These combined H_2_ and carbonaceous materials achieve the highest decarbonation levels in plastic waste treatment due to the sequestration of carbon in solid matrices that enter applications with a long lifespan. At the same time, they release respectable amounts of green hydrogen that can be used in different fields, such as ammonia production [88]. Its world demand is estimated to reach 650 MT in 2050, with more than 65% being produced using green hydrogen [89]. Green hydrogen will also feed (bio)refineries to produce the main petrochemicals. It is meant to play an equally crucial role in the decarbonation of metallurgy [90], in addition to its tremendous number of applications in energy production and industrial fields.

Despite the high potential of carbonaceous material production from plastic waste, these studies are still in the early stages of research and require more consolidation. The variety of techniques, feedstock, quality of obtained materials, and quality of target applications require more understanding of carbonization processes, the effects of feedstock quality and additives on the final products, and the limitations that can be induced on the properties of the final products. In fact, flaws in the structure of carbonaceous materials can have drastic effects on the target application when they are projected for industrial applications. More comprehensive studies correlating the types of feedstocks and the technologies used for the targeted applications are needed. On the other hand, the high energy demand and long processing times of the techniques used require the orientation of lab-scale studies from the perspective of industrial feasibility.

## 4. Possible Uses of Plastic Waste Pyrolysis Products in Industry

As mentioned earlier in this paper, pyrolysis produces three different phases of products: liquid, gaseous, and solid carbonaceous products. Based on feedstock and operating conditions, the repartition between these phases and their respective compositions can largely change. In the following section, we will discuss the energetic and non-energetic possible uses of these products.

### 4.1. Energetic Applications

Energetic applications consist mainly of using different cuts of pyrolytic oils as alternatives to fossil-based fuels. Although these uses reduce the volume of plastic waste, they contribute to global warming. When compared to fossil fuel lifecycle emissions, some reductions in the well-to-tank emissions can be expected, but they are still far from being considered low-carbon fuels given that the well-to-tank emissions of fossil fuels do not exceed 20% of their total emissions [91]. Still, energy applications can stand as a solution before advanced recycling solutions reach industrial maturity and economic viability. In this case, it could be a solution to waste management, taking into consideration the drawbacks of plastic degradation (microplastics) and spills into water courses. Furthermore, energy applications can also make sense if they avoid the extraction of new oil reserves; in this case, they should be integrated into the share of non-abated fossil fuels (and not added to it) in future energy mixes. Alternative fuel applications can also avoid all environmental drawbacks, other than the carbon footprint, related to fossil fuel extraction.

The literature is rich in papers dealing with the production, characterization, and utilization of plastic-based hydrocarbon cuts dedicated to transportation fuels. For this reason, this paper will focus on the non-energetic valorization of plastic waste.

### 4.2. Monomer Production

Petrochemical feedstock accounts for 14% of global oil demand and 8% of primary gas demand, and by 2050, according to the IEA Clean Technology Scenario, feedstock demand will reach 15.8 Mb/day, accounting for 26% of total oil demand, with a majority being dedicated to plastic production. The main monomers used are ethylene, propylene, styrene, and butadiene [92]. Ethane feedstock demand alone will grow by 70% by 2030 [93].

Light olefins can be obtained by separating the gaseous phase components of catalytic and thermal pyrolysis. Their production can also be optimized through the steam cracking of pyrolytic oils.

Besides polymers, petrochemicals enclose a wide variety of chemicals that stem from the same monomers as plastics in addition to methane.

### 4.3. Benzene, Toluene, and Xylene (BTX)

BTX (benzene, toluene, and xylene) are important chemical intermediates and building blocks of the chemical industry. They are used as solvents, precursors of medicines, and polymers, and they are also used in the printing industry and for leather tanning. They are among the most commonly produced petrochemicals, and their use is crucial in different industries. They are produced in refineries either by steam cracking or in reformers. According to Agilyx, the BTX feedstock represents 40% of total petrochemicals by volume and has a market value of USD 200 billion per year, which is expected to reach USD 500 billion by 2050 [94].

Recent studies show that BTX can be separated from the pyrolytic oils of aromatic-based polymers, but it can also be selectively produced from tires and polyolefins using appropriate catalysts and operating conditions [95,96].

### 4.4. Hydrocarbon-Based Lubricants

Industrial lubricants comprise a wide range of products depending on their application, such as simple soap greases, complex soup grease, grease with pigments, minerals, and polymers. Oil lubricants are the simplest. They are derived from crude oil distillation residues and are composed of isoparaffinic, naphthenic, and aromatic hydrocarbons [97]. They represent 1–2% of crude oil and have carbon chain lengths varying between 16 and 70 atoms [98].

Given the size and chain length of typical olefin- and paraffin-based lubricants, thermal cracking under mild temperature conditions can be suggested to obtain waxes that can be used in the production of lubricants. However, attention should be paid to the additives and impurities that mother polymers can contain.

### 4.5. Phase Change Materials

Hydrocarbons can also play a role in the production of phase change materials (PCMs), which are expected to be more popular in the future and will have a larger market revenue rate. In the year 2021, its market revenue was USD 1.66 billion, and it is expected to reach USD 5.1 billion by the year 2030. The current sources for PCMs are distributed among three categories: inorganic PCM (18%), organic PCM (36%), and bio-based PCM (46%) [99].

Phase change materials have the ability to exist in at least two phases (an amorphous one or more crystalline) and can switch between them while having electrical conductivity and thermal conductivity. All these properties give them the ability to store energy [100]. Among organic PCMs, hydrocarbons can cover a wide range of temperatures and enthalpies of melting. Kahwaji and White [101] investigated the temperature and enthalpy of the melting of alkanes with between 10 and 50 carbon atoms. They found the temperature of melting ranged from −25 to 100°C, and the phase change enthalpy ranged between 200 and 300 kJ/kg. Table 6 illustrates typical values of phase change temperatures and enthalpies of typical hydrocarbons, and it demonstrates the potential of using waxes from thermal cracking as phase change materials for low to mild temperature range applications.

### 4.6. Refrigerants

Concerns about global warming led the EU to phase out fluorinated gases used as refrigerants starting in 2030, with the first phase consisting of banning gases on new machines produced starting in 2025 and banning HFC in existing machines by 2030. Table 7 compares the global warming potential (GWP) of HFC to other alternatives, including HC-based fluids. As can be seen, pyrolytic gases from plastic waste can also be used as low-GWP alternatives to HFC.

## 5. Challenges and Future Prospects

### 5.1. Thermochemical Processing of Thermoplastics

As mentioned earlier, there are two ways to obtain target products: the catalytic pathway and thermal cracking, followed by upgrading.

With the catalytic pathway, the challenges are mainly related to the rapid deactivation of solid acid catalysts and the high cost of their production and regeneration. Furthermore, catalyst regeneration is mainly realized by oxidation using air or oxygen, where the exothermal nature of this reaction leads to the creation of hot spots inside the catalyst, leading to its destruction. To deal with these issues, research activities can focus on the following three aspects: (1) developing new, cheaper catalysts from clays or activated carbon resulting from pyrolysis activities; (2) working on soft recycling processes like ozonation, where regeneration can occur at temperatures as low as 100 °C; and (3) developing pyrolysis–reflux-coupled processes where the heavy products are recycled inside the pyrolysis reactor until reaching a carbon chain length compatible with the set reflux temperature.

When it comes to refining after pyrolysis, steam cracking, a petrochemical process that transforms hydrocarbons into shorter-chain olefins and aromatics, stands as the most promising solution as it has been practiced for the last two centuries on fossil oil. Soon, the existing petroleum refineries in the EU will drastically decrease due to mobility electrification by 2035, and the adaptation of plastic pyrolysis liquids will be of high economic interest. However, there are lots of challenges facing the application of such a combination. We can start by the amounts of collected plastics, with the largest plastic sorting facility in the EU barely reaching 100 kt/year, while the smallest steam cracking furnace requires 100 kt/year; a steam cracking facility is an assembly of several furnaces running in parallel [113]. At the same time, several studies have shown the limitations in mass and heat transfer induced by the low thermal conductivity and high viscosity of molten plastics inside the reactor [114]. These limitations lead to a maximum-size pyrolysis reactor that can deal with 15–20 kt of plastic per year [22]. Secondly, steam cracking furnaces are designed to run with a stable composition of feedstock with a precise range of physical and chemical characteristics that does not have too much tolerance to additives and heteroatoms that plastics and their pyrolytic oils can contain, without mentioning the seasonal variability of plastic waste feed [115].

Thus, research should focus on (1) hydrotreating for heteroatom removal by selecting cheap catalysts and optimizing conditions in order to lower the hydrogen consumption of the plant; (2) demineralizing using eco-friendly techniques by choosing appropriate extraction solvents; (3) working on multiphysics models and developing digital twins that could help in downsizing the steam cracking process and increasing the size limit of pyrolysis reactors; and (4)cofeeding pyrolytic oils with fossil feedstock on a first time-lapse and then intensifying research on cofeeding pyrolytic plastic oils with pyrolytic oils from biomass-based wastes.

### 5.2. Thermochemical Processing of Fiber-Filled/Reinforced Plastics

Plastic may often contain reinforcing fillers or chopped fibers that increase the polymer’s mechanical performance (e.g., stiffness and strength). In particular, continuous fiber-reinforced polymer composites (FRPs) are widespread in lightweight, high-performance structures. There is a growing concern regarding their sustainability, notably at the end of life, as only 2% of end-of-life FRP is currently recycled worldwide [116]. One of the major recycling challenges is the variety of materials that constitute FRPs because different polymer matrices (various thermoplastic or thermoset resins) and different reinforcement fibers (mainly glass or carbon fibers) require different treatments.

Different recycling technologies can be applied to polymer composites: mechanical recycling; thermal processes such as pyrolysis, steam or fluidized-bed thermolysis, and microwave-assisted pyrolysis; and solvolysis at high or low temperatures and pressures. The recycling technique must be selected according to the material being recycled and also the reuse application (i.e., recovery and reuse of valuable products from the polymer fraction or the fibrous fraction) [117]. Whereas mechanical recycling generates recyclates that generally have very low value, thermochemical processes, such as solvolysis and pyrolysis, may be considered “advanced recycling“ processes that enable the recovery of higher-value recyclates, such as intact fibers and sometimes matrix degradation products [118]. The technology readiness level (TRL) assessment of these composite recycling technologies has highlighted a lack of industrial maturity of the thermochemical processes compared to mechanical recycling and the need to understand the use of reclaimed materials as an important assessment element of recycling efforts [119]. As a matter of fact, most of the research efforts targeting the separation of fibers from the polymer matrix have focused on the recovery and reuse of the fibers (especially carbon fibers, which are highly valuable [120], and glass fibers to a lesser extent because of the lack of economic viability); relatively few studies have examined the recycling/recovery/valorization of the polymer phase (e.g., recovering chemical products from the polymer matrix recyclates).

Thermochemical processes are not yet viable for glass fiber-reinforced polymer composites considering the low price of virgin glass fibers and the degraded mechanical performance of glass fibers recovered in this way; they may, however, be used to recover valuable products (e.g., carboxylic acids, glycols, and styrene–fumaric acid copolymer) from the organic phase (e.g., unsaturated polyester matrix) to reuse as monomers or additives in new resins [117]. On the other hand, thermochemical processes are suitable for carbon fiber-reinforced polymer composite recycling due to the high value of the carbon fibers; pyrolysis is commercially exploited and some solvolysis processes are available for commercial exploitation. Shorter carbon fiber fractions can be reused in sheet or bulk molding compounds in place of virgin carbon fibers. Longer fiber fractions can be reused in more structural applications, providing a re-alignment of the recovered carbon fibers. Spinning techniques also offer interesting perspectives for reshaping the reclaimed carbon fibers into continuous yarns [117]. Another challenge is to maintain the added value of the woven architectures of woven fabrics; thermochemical treatments have shown their ability to retain the woven shape of the reinforcement fabrics. Developing ready-to-use semi-products (sheet or bulk molding compounds, laminates, or preforms) could be the key to unlocking the re-incorporation of reclaimed carbon fibers into second-life composites [117,118].

Although the organic fractions recovered after thermochemical separation have received little attention, a recent review of progress on the recovery of chemical recyclates from fiber-reinforced composites has comprehensively addressed the solvolysis of different families of composite materials made of carbon or glass fibers and thermoset matrices (amine-epoxy, anhydride-epoxy, and unsaturated polyester) or thermoplastic matrices (elium polyacrylates; polyamides; and some commodity polymers, such as PET, HDPE, and PP) [118]. It is worth noting that although there are some parallels between the engineering thermoplastics used as composite matrices and commodity thermoplastics, the presence of fibers and the higher Tg of these materials make their solvolysis much more tricky.

## 6. Conclusions

Only 9% of the plastics produced cumulatively worldwide since 1950 have been recycled. Actually, recycling rates do not exceed 30% and mainly rely on mechanical processing. The complexity of these materials and their composition leads to a costly sorting process with less than 50% sorting rate. The use of chemical processes is very promising and would greatly increase recycling rates. Pyrolysis and gasification are among the most advanced processes and are being applied on an industrial scale. Most of the existing applications are oriented towards fuel/energy recovery, which raises the dilemma of reducing waste at the expense of increasing the carbon footprint of plastics. Using these processes in order to produce chemical feedstock is a good compromise, leading to simultaneous waste and carbon footprint reduction along with creating added-value products. The variety of operating conditions, catalysts, and processes leads to open choices in terms of targeted final products, such as linear or branched hydrocarbons ranging from light gases to heavy paraffins, aromatics, syngas, hydrogen gas, and carbonaceous materials. These final products can be used in a wide variety of non-energetic applications, such as the production of monomers, which can close the loop of plastics; intermediate hydrocarbons, which can serve as solvents, lubricants, phase change materials, or even refrigerants; syngas, which can be used in Fischer–Tropsch process to produce new feedstock hydrocarbons; and hydrogen, which can be used for a multitude of applications. Besides that, carbonaceous materials can cover a wide variety of applications in catalysis, water treatment, energy storage, light and resistant composite materials, etc. Still, lots of challenges are facing these applications, such as the economic viability related to feedstock dispersion, feed variability, separation and purification of final products and byproducts, catalyst regeneration, and mass and heat transfer within processes related to the nature of the plastic (high viscosity and low thermal conductivity).

## Figures and Tables

**Figure 1 materials-17-01460-f001:**
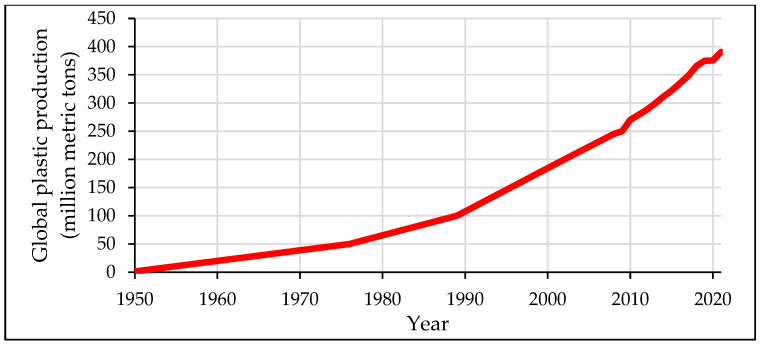
Global plastic production in million metric tons from the year 1950 to 2021 [2].

**Figure 2 materials-17-01460-f002:**
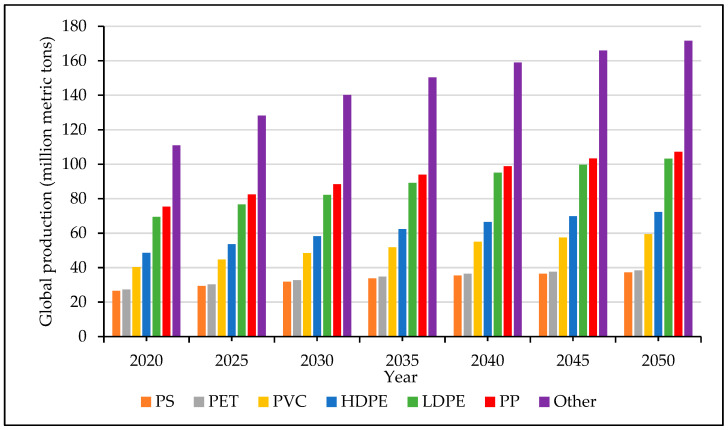
Global thermoplastic production forecast per group in million metric tons from the year 2020 to 2050 [4].

**Figure 4 materials-17-01460-f004:**
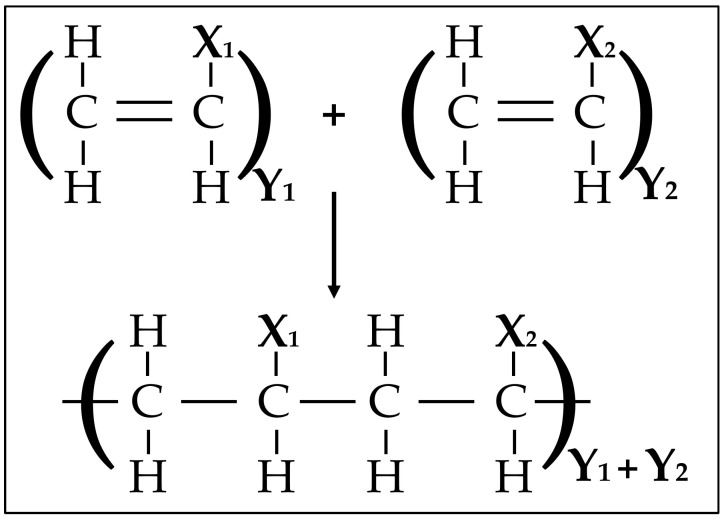
Addition polymerization (radical, cationic, or anionic) chemical reaction [35].

**Figure 5 materials-17-01460-f005:**
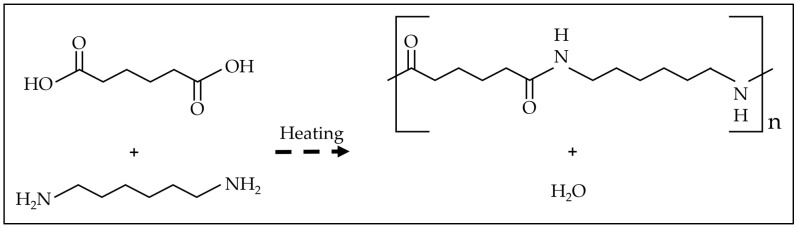
Synthesis of nylon, which is an example of condensation polymerization of di-carboxylic acid and 1,6-hexamethylenediamine [37,38].

**Table 1 materials-17-01460-t001:** Some additives found in plastic waste [13,14,15,16,17].

Additive Type	Examples
Plasticizers	Benzyl butyl phthalate
	Di-isoheptyl phthalate
	Di-isobutyl phthalate
	Dibutyl phthalate
	Bis (2-ethylhexyl) phthalate
	Bis(2-methoxyethyl) phthalate
	Tris(2-chloroethyl) phosphate
Stabilizers	Arsenic compounds
	Triclosan
	Organic tin compounds
	Bisphenol A (BPA)
	Octylphenol
	Cadmium compounds
Colorant	Titanium dioxide
	Cobalt (II) diacetate
	Chromium compounds
Curing agents	Formaldehyde
	4,4′-Diaminodiphenylmethane
	2,2′-dichloro-4,4′-methylenedianiline

**Table 2 materials-17-01460-t002:** Common miscible polymer blends and their improved characteristics.

	Blend	Properties	Ref.
Homopolymer–homopolymer	Polyphenylene oxide (PPO)–polystyrene (PS)	Known as NOYRL^TM^, may be designed to replace metallic parts in mechanical assemblies	[23]
	Polyethylene terephthalate (PET)–polybutylene terephthalate (PBT)	Heat and chemical resistance, along with excellent processability	[24]
	poly(methyl methacrylate) (PMMA)–polyvinylidene fluoride (PVDF)	Combines the rigidity of PMMA and the flexibility of PVDF while lowering the melting point	[25]
Homopolymer–copolymer	Polypropylene (PP)–EPDM rubber	Increased tensile strength	[26]
	Polycarbonate (PC)–acrylonitrile butadiene styrene (ABS)	Improved toughness, processability, and thermal stability	[27,28]

**Table 3 materials-17-01460-t003:** Hydrolysis process of polymers.

Polymer/Feedstock	Type	Operating Conditions	Products	Ref.
PET	Acid hydrolysis	Depolymerization of PET occurs at 100 °C in 96 h and may add catalysts, such as MSO_4_	Terephthalic acid (TPA) and ethylene glycol (EG)	[41]
PET	Alkaline hydrolysis	Carried out in an alkaline solution of NaOH or KOH with a concentration range of 4–20 wt%, with the reaction taking 3–5 h at 210–250 °C	Ethylene glycol (EG) and terephthalic salts	[42]
PET	Transesterification (methanol)	Depolymerized at 65 °C in a stirred reactor, catalyzed using Na_2_CO_3_ and a MeOH/EG molar ratio > 15 for 90 min	DMT + EG	[43]
Polylactic acid (PLA)	Hydrolysis	Carried out at a temperature of 60–80 °C and takes 15–50 days	Lactic acid	[44]
Nylon polymers (nylon 6, nylon 6/6, nylon 12, and nylon 6/12)	Acid hydrolysis	Carried out at 350 °C for 10 min	ϵ-caprolactam	[45]

**Table 4 materials-17-01460-t004:** The pyrolysis proces of polymers.

Polymer/Feedstock	Type	Operating Conditions	Products	Ref.
LDPE	High-pressure pyrolysis	High pressure up to 51 bar, initial temperature of 330–380 °C, exceed the set temperature by 100 °C at a rate of 150 °C/min, stirring at 200 rpm	Aromatic compounds, isoparaffins, and cycloalkanes	[56]
PET	Catalytic pyrolysis	Heating at 700 °C in the presence of a Ca(OH)_2_ catalyst	Benzene (used for lubricants, dyes, and detergents)	[57]
HDPE	Catalytic cold plasma pyrolysis	18 h at room temperature, then calcinated to 500 °C for 6 h in the presence of HZSM-5 zeolite or sulfated zirconia catalyst	Ethylene	[58]
HDPE, LDPE, PP, PS	Catalytic pyrolysis	Stirring at 50 °C until the mixture becomes slurry, then heating at 105 °C for 12 h and calcination at 800 °C for 2 h at a rate of 20 °C/min in the presence of an Fe/Al_2_O_3_ catalyst	Amorphous carbon, carbon nanotubes, and hydrogen	[59]
PP	Catalytic pyrolysis	Stirring at 100 °C, then drying in an oven at 120 °C and calcination at atmospheric pressure at 500 °C using Ni-Cu/La_2_O_3_ catalyst	Multiwalled carbon nanotubes and carbon nanofibers	[60]
PS	Flash pyrolysis	Operating at 500 °C	Styrene with byproducts (toluene, ethyl benzene, and α-methyl styrene)	[61]
Single-use face masks (PP)/food waste	Single-step pyrolysis	Operating at 900 °C	Hydrocarbon mixtures and hydrogen	[62]

**Table 5 materials-17-01460-t005:** Selected plastic pyrolysis plants in Europe and North America [67,68,69,70,71].

Technology Provider	Capacity in Tons per Day	Feedstock	Products	Technology Utilized	Location
Alterra energy	60	HDPE, LDPE, PP, PS, and “other” types of plastics	Syncrude and diesel	Rotary kiln	Akron, OH, USA
Nexus	50	HDPE, LDPE, PP, and PS with contamination ≤ 1% PVC and ≤2% PET	Light crude, diesel, gasoline, kerosene blendstocks, and wax	Melting vessel	Atlanta, GA, USA
Agilyx	10–50	Film HDPE, LDPE, PP, and PS	Light synthetic crude oil	Dual screw reactor	Tigard, OR, USA
Recycling Technologies	20	Soft and flexible packaging (films), multilayered and laminated plastics (crisp packets), and complex and contaminated plastics (food trays)	Low-sulfur hydrocarbon Plaxx—wax	Fluidized bed	Swindon, UK
Plastic Energy	20–30	Rigid and film HDPE, LDPE, PP, and PS	Raw diesel, light oil, and synthetic gas components	Stirred-tank reactor	Sevilla, Spain
Susteen Technologies	12	Mainly residual biomass and sewage sludge	Green crude, diesel, gasoline, and jet fuel	Screw with recirculation	Sulzbach-Rosenberg, Germany
PHJK	12–14	Unsorted plastic waste	Light crude oil and diesel	Rotary kiln	Laihia, Finland

**Table 6 materials-17-01460-t006:** Common hydrocarbon phase change materials [102,103,104,105,106,107,108].

PCM	Melting Point (°C)	Latent Heat (kJ/kg)	Thermal Conductivity (W/m.K)
n-Tetradecane	6	228–230	0.14
n-Pentadecane	10	205	0.2
n-Hexadecane	18	237	0.2
n-Heptadecane	22	213	0.145
n-Octadecane	28	245	0.148
n-Docosane	44.5	249	0.2

**Table 7 materials-17-01460-t007:** Global warming potential (GWP) of common fluorinated [109] and alternative gases [110,111,112].

Gas	GWP (AR4^9^, 100 Year)
CO_2_	1
Ammonia (NH_3_)	0
Nitrous oxide	298
Hydrocarbons	
Methane	25
Propane (R-290)	3
Isobutane (R-600a)	3
Propylene (R-1270)	<1
HFC	
R134a	1430
R407C	1774
R410A	2088
R404A	3922
HFC-125	3500
PFC-14	7390
SF_6_	22,800

## Data Availability

Data are contained within the article.

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
