# Peer review of "An Overview of the Non-Energetic Valorization Possibilities of Plastic Waste via Thermochemical Processes"

_materials, 2024, doi:10.3390/ma17071460_

Round 1

Reviewer 1 Report

Comments and Suggestions for Authors

The submitted article entitled „An overview on Non-Energetic Valorization Possibilities of 2

Plastic Waste via Thermochemical Processes” is a well-written article about a hot-topic area.

1.       Abstract: Abstract needs to be reworded. Currently it is like a "mini-introduction" with an aim at the end. Instead, the abstract should summarise the topic indicated in the title in 5-8 sentences (without unnecessary fragmentation). E.g. the first 6 sentences in this section are needless, these facts will be discussed later in the actual Introduction. The first sentence is misleading anyway; the authors probably only mean synthetic polymers by "polymeric materials"

2.       Introduction: The introduction has a good and logical approach.

3.       Chapter 2: this section is a valuable part of the manuscript, although it is not closely related to the title's theme, but can be seen as a continuation of the introduction. For this reason, this section should be abridged and merged with the first (Introduction) chapter. Figures 2 and 3 are not necessary, they are only marginally related to the topic and do not help to understand the main message.

4.       Chapter 3: comprehensive part, this should follow the merged Introduction

5.       Chapter 4: 4.1: which types of plastics can be recovered by this method?  4.2: this part almost excludes the term polymerisation instead of depolymerisation. For this reason it should be rewritten.  Table 5.: the table could be completed with information on which plastics can be processed by which processes (e.g. type or origin)

6.       Conclusions: the first three sentences are redundant here, they do not state a conclusion.

Minor comments:

"Figure" instead of "figure";

title: "Overview" instead of "Oveview"

In summary, the manuscript is of good quality, but sometimes overly voluminous and diverges from the given topic.  This can be corrected by omissions and mergers in some places. In other places, however, minor additions would help to improve the technical content. Overall, the manuscript contains valuable results, which I recommend for publication after major revision.

Comments on the Quality of English Language

The English language is adequate, only minor corrections are needed.

Author Response

We thank the reviewers for their helpful and constructive comments.  In the uploaded word file we refer to the reviewer’s comments one by one; our responses appear in green. The modified text in the manuscript is highlighted in yellow.

Reviewer 2 Report

Comments and Suggestions for Authors

The review article “An oveview on Non-Energetic Valorization Possibilities of Plastic Waste via Thermochemical Processes” presents a review to detail the liquid and gaseous hydrocarbons with a specific focus on plastic materials waste thermal degradation.

Some corrections should be addressed before publication.

Check the title and correct it.

L72. Add more information to the sentence 'break down complex polymeric structures' What are those polymeric structures? In addition, the authors must explain the benefits of mechanical recycling in commodity polymers, which are actually the most consumed worldwide.

It is recommended not to diminish the advantages of mechanical recycling but to highlight the properties of chemical recycling.

The authors should also highlight the harmful effects of chemical recycling, such as the cost and pollution from using solvents and reactors.

L116. Figure 2. Please provide the reference.

L147. Table 1. The authors must enrich the table with more references since this is a review article, and it is not acceptable for Table 1 to be built from a single reference.

L211. Figure 5 and Figure 6. Please provide a reference.

L443-444 5.3. the title should be Benzene, Toluene and Xylene instead of BTX. Change the format.

L483. Table 6 and Table 7. Common hydrocarbon phase change materials [75]. The authors must enrich the table with more references since this is a review article, and it is not acceptable for Table 1 to be built from a single reference.

Author Response

We thank the reviewers for their helpful and constructive comments. In the uploaded document we refer to the reviewer’s comments one by one; our responses appear in green. The modified text in the manuscript is highlighted in yellow.

Round 2

Reviewer 1 Report

Comments and Suggestions for Authors

The corrections made are appropriate and I recommend that the manuscript be accepted.